# Exosomes as a Nano-Carrier for Chemotherapeutics: A New Era of Oncology

**DOI:** 10.3390/cells12172144

**Published:** 2023-08-25

**Authors:** Rodrigo Pinheiro Araldi, Denis Adrián Delvalle, Vitor Rodrigues da Costa, Anderson Lucas Alievi, Michelli Ramires Teixeira, João Rafael Dias Pinto, Irina Kerkis

**Affiliations:** 1Genetics Laboratory, Butantan Institute, São Paulo 05503-900, SP, Brazil; denisdelvalle05@gmail.com (D.A.D.); vitor_rodriguesdacosta@hotmail.com (V.R.d.C.); andersonlucasalievi@gmail.com (A.L.A.); michelli.teixeira@butantan.gov.br (M.R.T.); 2Structural and Functional Biology Post-Graduation Program, Paulista School of Medicine, São Paulo Federal University (EPM-UNIFESP), São Paulo 04023-062, SP, Brazil; 3BioDecision Analytics Ltd.a., São Paulo 13271-650, SP, Brazil; joao.dias@biodecisionanalytics.com; 4Endocrinology and Metabology Post-Graduation Program, Paulista School of Medicine, São Paulo Federal University (EPM-UNIFESP), São Paulo 04023-062, SP, Brazil

**Keywords:** extracellular vesicles, exosomes (Exo), nano-delivery, cancer, mesenchymal stem cell (MSC), MSC-Exo

## Abstract

Despite the considerable advancements in oncology, cancer remains one of the leading causes of death worldwide. Drug resistance mechanisms acquired by cancer cells and inefficient drug delivery limit the therapeutic efficacy of available chemotherapeutics drugs. However, studies have demonstrated that nano-drug carriers (NDCs) can overcome these limitations. In this sense, exosomes emerge as potential candidates for NDCs. This is because exosomes have better organotropism, homing capacity, cellular uptake, and cargo release ability than synthetic NDCs. In addition, exosomes can serve as NDCs for both hydrophilic and hydrophobic chemotherapeutic drugs. Thus, this review aimed to summarize the latest advances in cell-free therapy, describing how the exosomes can contribute to each step of the carcinogenesis process and discussing how these nanosized vesicles could be explored as nano-drug carriers for chemotherapeutics.

## 1. Global Burden of Cancer Metastasis

Cancer is a major public health problem and the second leading cause of death worldwide [1], causing 10 million deaths globally in 2022 alone [2].

According to GLOBOCAN, about 19.3 million new cases of cancer were registered globally in 2020 [3], representing about 5250 new cancer cases each day in the United States alone for the same year [4]. The tabular data and graphical visualization of the historical series of cancer can be accessed via the Global Cancer Observatory (GCO) (http://gco.iarc.fr, accessed on 13 March 2023). However, the epidemiological projections for cancer are alarming. Data from the Global Cancer Observatory indicate that 28.8 million new cancer cases and 16.1 million deaths by the disease will occur in 2040 [5]. Following these projections, it is estimated that there will be 34 million new cancer cases, twice the number estimated in 2018 [2]. These numbers emphasize the urgent need for innovative strategies to prevent and treat cancer [6].

Cancer is a multifactorial disease. For this reason, it is not surprising that there are different causes of this increasing cancer incidence and mortality. Among these factors, aging remains the most important risk factor for cancer [7]. This is because there are several known mechanistic drivers (i.e., cumulative mutations) or aging hallmarks (i.e., genomic instability, (epi)genetic alterations, chronic inflammation, metabolic dysregulation, and dysbiosis) that stimulate oncogenesis and, subsequently, cancer progression and metastasis [7].

Due to the complexity of the oncogenic process, there not yet a unified explanation for its cause [8]. In an attempt to conceptualize the vast complexity of the oncogenic process, Hanahan and Weinberg proposed a set of functional capabilities acquired by normal cells that drive them to become cancer cells, which are known as hallmarks of cancer [9,10].

According to these hallmarks, cancer is caused by mutations in genes that regulate the cell cycle (oncogenes or tumor suppressor genes), conferring malignant properties to normal cells [8]. However, these mutations also drive a wave of cellular multiplication associated with a gradual increase in tumor size, disorganization, and malignancy [10,11,12]. For this reason, Hanahan [13] summarized the multistep process of carcinogenesis in eight hallmarks that comprise (i) the acquired capability to sustain proliferative signaling, (ii) evading growth suppressors, (iii) resisting cell death, (iv) enabling replicative immortality, (v) inducing vasculature, (vi) activating invasion and metastasis, (vii) reprogramming cellular metabolism, and (viii) avoiding immune destruction.

However, due to the multistep nature of the oncogenic process, cancer generally is a “silent” disease. Therefore, although the early diagnosis of cancer is crucial for successful treatment, the lack of signals and symptoms at the beginning of the disease means that many diagnoses occur when the patient exhibits local or distant dissemination, negatively impacting the survival rate. Thus, it is not surprising that 90% of cancer deaths are associated with cancer metastasis [14].

Cancer metastasis remains challenging given the available therapeutic approaches. Nonetheless, the cumulative information about the molecular and biochemical mechanisms that govern this process generated in the past two decades opens up novel opportunities to provide more appropriate therapeutic approaches for patients with metastasis [15,16,17,18]. Based on this, herein we aimed to summarize the main metastasis-related mechanisms, identifying therapeutic opportunities from which the extracellular vesicles (EVs) can be biotechnologically explored as novel therapeutics, thereby serving as vehicles for drug nano-delivery.

## 2. Metastatic Process

Metastasis is a multistep process that includes (i) local infiltration of tumor cells into adjacent tissues, (ii) transendothelial migration of cancer cells into vessels (intravasation), (iii) survival in the circulatory system, (iv) extravasation, and (v) proliferation in competent organs leading to colonization [19]. However, for each step, there are a multitude of (epi)genetics, biochemical, and morphological alterations involved [16,20]. Thus, given the complex nature of the metastatic process, which requires a coordination of events, it is not surprising that metastasis is an inefficient process [19]. In this sense, although a primary tumor having a size of 1 cm (about 1 × 10^9^ cells) can disseminate one million cancer cells per day [21], less than 0.1% of disseminated cancer cells successfully develop a distal metastasis [19]. Despite the small number of disseminated cancer cells that successfully colonize distant organs, metastasis remains the main cause of death by cancer globally [14]. This is because the genetic and metabolic deregulations verified in metastatic cells confer important adaptative advantages to these cells.

Although cell invasion is the first step of the metastatic process, the genetic and metabolic deregulations that enable cell invasion occur during cancer progression [20,22]. This is because the cell growth and proliferation verified during the cancer progression naturally limit the supply of oxygen and nutrients to cancer cells, creating hypoxic zones [19,22]. These zones arise as a consequence of an imbalance between oxygen supply and consumption in solid malignant tumors [19,22]. These zones are characterized by oxygen pressure between 0 and 20 mmHg (1–2% or below). In normal healthy tissues, the oxygen tension is nearly 40 mmHg (about 5%) [19].

Hypoxia (which is a common feature of solid tumors) activates hypoxia inducible factor 1 alpha (HIF-1α), promoting the expression of numerous genes whose products coordinate the adaptative response [23,24].

HIF-1α is a cytoplasmic protein regulated by oxygen levels and plays a key role in modulation of tumor energy metabolism [24,25,26,27]. When the oxygen supply is sufficient, HIF-1α is hydroxylated at proline residues through oxygen-dependent enzymes. Once hydroxylated, the prolyl sites of HIF-1α binds to von Hippel Lindau tumor suppresser (pVHL) [23,28,29]. Under hypoxic conditions, non-hydroxylated HIF-1α is translocated to the nucleus, where it binds to HIFβ (a nuclear subunit, constitutively expressed and independent of hypoxic conditions) to form heterodimers [23,30]. These heterodimers bind to target genes, promoting energy metabolic alterations, aberrant angiogenesis, cell growth, and metastasis [23,30].

Cancer demands ATP to fulfill the biosynthesis needed for proliferation [26]. Under physiological conditions, normal cells produce ATP via the oxidative metabolism. Using a series of coordinated enzymatic reactions, these cells convert the glucose in pyruvate in a cytoplasmic reaction (glycolysis) that occurs independently of oxygen. In the presence of oxygen, pyruvate is converted to acetylcoenzyme A (acetyl-CoA), which enters the tricarboxyclic acid cycle (TCA, also known as the Krebs cycle) [20]. In turn, this generates mitochondrial membrane potential (ΔΨm) [20,31]. Mitochondrial membrane potential promotes proton transport from the inter-membrane space of the mitochondrial membrane, which is necessary for ATP synthesis [31]. ATP is synthetized by ATP synthase via ADP linking to inorganic phosphate (Pi) [31]. However, this process results in ROS production [20,32]. Thus, in normoxic conditions, each mol of glucose produces about 36 mols of ATP. However, under hypoxic conditions, this energy balance is committed. In an attempt to guarantee the energy supply to cancer cells, HIF-1α activation drives the metabolism for a glycolytic pathway [20]. However, in contrast to the oxidative metabolism, the anaerobic metabolism produces only two mols of ATP per mol of glucose. To solve this problem, HIF-1α activates the expression of glucose transporters (GLUTs), increasing the glucose uptake in order to produce sufficient amounts of ATP to meet the cancer energy demand [20,33]. However, the cancer cells’ metabolism leads to the accumulation of lactate (or lactic acid) in the cytosol, which, together with the proton hydrogen (H+), must be released into the extracellular space to prevent intracellular acidification [34]. For this reason, cancer cells export both lactate and H+ into the tumor microenvironment (TME) through monocarboxylate transporters (MCTs) and Na-driven proton extrusion [34,35].

MCTs are members of the solute carrier transporter family (SLC) [34]. The family SLC16 encodes fourteen MCT isoforms, from which only four isoforms (MCT1-4) serve as lactate transporters [34]. MCT1 is responsible for both lactate and pyruvate upload, whereas MCT4 exclusively exports lactate and H+ [34].

Lactate is one of the most abundant metabolites found in the TME [34,35]. Under physiological conditions, lactate varies in a concentration range from 1.5 (blood) to 3 mM (healthy tissues) [10]. However, within the TME, lactate concentration is commonly observed to be higher than 40 mM [36]. Thus, while under physiological conditions the pH of blood and tissues is controlled at around 7.4, in the TME the local pH typically ranges from 5.6–7.0 [37]. The TME acidification creates a permissive microenvironment for many of the acquired characteristics of cancer cells, facilitating tumor immune escape and effective proteolytic degradation of the extracellular matrix (ECM) by invading cancer cells [34,35].

Thus, lactate serves as a key molecule in the immune escape, which is recognized as one of the a hallmarks of cancer [10]. The TME is a heterogeneous environment, composed of cancer cells, cancer-associated fibroblasts (CAFs), tumor endothelial cells, and immune cells from both innate (macrophages, neutrophils, dendritic cells, innate lymphoid cells, myeloid-derived suppressor cells, and natural killer cells) and adaptative systems (T and B cells) [34]. For this reason, it is not surprising that cancer cells exhibit a mechanism of immune escape to guarantee their survival.

In this sense, it was demonstrated that concentrations of lactate higher than 20 mM induce apoptosis of natural killer cells (which exhibit antitumoral activity) [38,39]. Lactate also has the following effects: (i) Lactate inhibits T cell proliferation, altering cytokine production [40,41]. (ii) Lactate prevents dendritic cells’ differentiation (antigen-presenting cells that a play role in immune responses), leading to the production of interleukin (IL)-10 (animmuno-suppressive cytokine) [34]. (iii) The secretion of IL-10 within the TME inhibits the production of proinflammatory cytokines such as interferon (IFN)-γ, tumor necrosis factor (TNF)-α, IL-1β, and IL-6 [34]. (iv) Lactate also promotes the development of myeloid-derived suppressor cells (MDSCs), a heterogeneous population of immature myeloid cells that suppress both innate and adaptative immunity by preventing the maturation of dendritic cells [34,42]. Altogether, these actions lead to immune escape.

Beside this, lactate facilitates the cancer cells’ invasion and migration. This is because lactate promotes the activation of metaloproteinases (MMPs), proteases that belong to the superfamily of zinc-endopeptidases, which display a specific proteolytic activity against numerous substrates located on the ECM [43,44,45].

In addition, HIF-1α activation promotes the upregulation of pro-angiogenic proteins, such as vascular-endothelial growth factor (VEGF), promoting the formation of new blood vessels [46]. In this regard, the ECM degradation is crucial to the formation of novel blood vessels (neo-angiogenesis) and, therefore, facilitates cancer cell dissemination.

However, even after angiogenesis, when the oxygen supply is restored, cancer cells preferentially use glycolysis rather than oxidative phosphorylation for energy production [20,43,47,48,49]. This interesting phenomenon was described a century ago by Otto Warburg and is known as “Warburg effect” [48,50,51].

The Warburg effect is an important metabolic reprogramming process to ensure the survival of cancer cells. This is because, to provide sufficient ATP to sustain the biosynthetic process verified during the oncogenesis using the oxidative metabolism, mitochondria will produce high levels of reactive oxygen species (ROS), such as super oxide radicals (O_2_^−^) and hydrogen peroxide (H_2_O_2_), which could cause cell death [20].

Furthermore, once activated, HIF-1α binds to epithelial-mesenchymal transition (EMT)-related transcription factors, such as Snail1 (Snail), Snail2 (Slug), ZEB1/2, and TWIST [16,52,53].

EMT is a biological reprogramming process (transdifferentiation), characterized for multiple coordinated events in which epithelial cells acquire a mesenchymal-like phenotype, conferring migratory and invasiveness ability associated with anoikis resistance [20,54,55,56,57,58].

EMT was first reported in 1908 by Frank Lillie, and later described by Elizabeth Hay in corneal epithelial tridimensional (3D) cell cultures [20,59,60,61]. EMT is a natural process that occurs during embryogenesis (EMT type 1) [62,63,64] or in adult life during the wound healing process (EMT type 2) [65]. However, inappropriate activation of EMT can cause important disturbances of epithelial tissue homeostasis and integrity, which are associated with several diseases, including cancer (EMT type 3) [55].

EMT is a reversible transdifferentiation program regulated by the nuclear transcription factors (Snail, Snail2, ZEB1/2, and TWIST) and able to bind to the E-box region of *CDH1* and *CDH2* genes, which encode E- and N-cadherin, respectively [57,66,67]. These transcription factors promote E-cadherin downregulation (leading to the loss of cell-to-cell adhesion) and N-cadherin upregulation (resulting in the formation of transient sites of adhesion necessary for cell migration) [20,68].

E-cadherin comprises the adhesion junctions, together with the cytoskeleton and α-, β-, and γ-catenins, and is crucial for the maintenance of the epithelial phenotype [68]. Loss of E-cadherin in cancer cells leads to metastatic dissemination and activation of several EMT transcription factors [68]. This is because the destabilization of adherens junctions leads to the release of β-catenin, which acts as a transcriptional factor [68]. β-catenin upregulates the Signal Transducers and Activators of Transcription (STAT)-3, which was identified as the major contributor to the formation of cancer stem cells (CSCs) or tumor initiating cells (TICs) [16,69,70,71,72].

CSCs comprise a small population of cancer cells, originated from either differentiated cells or adult tissue resident stem cells [73,74,75,76]. These cells have been known since 1877, when Virchow’s student Cohnhein reported the presence of a cell population possessing an embryonic character within the TME [76]. Although the presence of these cells within the TME was first reported in 1877, CSCs were identified in leukemia in the 1990s [77,78,79].

However, these cells only gained attention only in the past few decades due to their ability to drive tumor initiation and can cause relapses [80].

Because they share many characteristics with stem cells, such as the self-renew capability and differential potential conferred by the expression of pluripotency markers (OCT-3/4, Sox2, Nanog, KLF4, and MYC) which are involved in recurrence, metastasis, heterogeneity, and heterogeneity, these cells were named cancer stem cells [79]. In addition, as well as being observed in stem cell populations, CSCs also express high levels of ATP-binding cassette (ABC) transporters [81] and DNA damage response genes [79,82]. The overexpression of these genes increases the efflux of chemotherapeutics and reduces the radiosensitivity, conferring mechanisms of chemo- and radioresistance, thereby explaining the high mortality rate in patients with cancer metastasis [83,84,85,86].

Cancer cell plasticity verified during the EMT not only leads to the formation of CSCs, but also promotes morphological alterations, which are crucial to ensuring the escape of primary tumor. Thus, the cell plasticity verified during the oncogenic process is a natural cell reprogramming process activated to guarantee the cell survival in response to the oxygen- and nutrient-starving conditions of the TME [73,74,87].

As consequence of this cell plasticity, cancer cells acquire a fibroblastic-like form [88,89]. This phenotype facilitates the passage of cancer cells cross the endothelial barrier, resulting in the intravasation of both blood and lymphatic vessels [90]. After the intravasation, cancer cells are transported in the circulation [91,92,93].

However, the activity of the immune system within the circulatory system plays an important role in suppressing tumor progression [93]. For this reason, cancer cells exhibit different strategies to protect themselves against immunological and mechanical stress [93]. In this context, the cancer cell extravasation can be considered a mechanical strategy to escape the TME [93,94].

The extravasation of cancer cells preferentially occurs in small capillaries [93]. Thus, cancer cells can (i) directly transmigrate the endothelium as single cells or (ii) extravasate through mechanisms that involve adhesion to the endothelium, modulation of the endothelium barrier, and transendothelial migration [93,94]. Both transmigration and extravasation enable cancer cells to reach the underlying tissues [93,94].

However, to reach and colonize the underlying tissues, it is necessary that the metastatic niche is supportive and permissive of the colonization of cancer cells [95]. In this sense, studies have been shown that cancer-derived EVs (particularly exosomes/oncosomes) mediate the formation of pre-metastatic niches, creating a permissive environment to support the metastatic cell colonization [95,96,97]. Given the importance of EVs in the metastatic process, the biology of these vesicles is discussed in the following sections.

## 3. A Brief History of Extracellular Vesicles

Extracellular vesicles (EVs) is a generic term used to refer to a heterogeneous group of small cell-released particles with a diameter ranging from 10 to 1000 nm [98].

Although the interest in these vesicles emerged two decades ago, the EVs’ history started in the second half of the 1940s. In 1945, while studying blood coagulation, Chargaff observed small “membrane debris” sediments following high-speed centrifugation of plasma supernatant [99]. The presence of this “debris” was confirmed in another study published by Chargaff one year after his first report, describing “a variety of minute breakdown products of blood corpuscles” [100]. Twenty years later, using electron microscopy, Peter Wolf confirmed the existence of corpuscles “originated from platelets”, which were described as “platelet dust” [101], supporting Chargaff’s studies [100]. Despite these publications, the biological nature of these corpuscles remained unknown until 1974, when Nunez et al. [102] identified structures with a size under 1000 nm, called multivesicular bodies (MVBs), opening a path for the identification of a subtype of EVs that later was called exosomes or small EVs (30–150 nm) [103].

Basically, EVs are classified based on their biogenesis mechanisms in exosomes, microvesicles (MVs), and apoptotic bodies [75,104,105]. However, currently, new terminologies are used to classify these vesicles. Thus, the EVs can be also classified based on their concept (e.g., oncosomes, matrix vesicles, stress EVs, and migrasome) and size (e.g., small EVs and larges EV) [104], as described in Table 1.

Among these EVs, exosomes are the most studied class of extracellular vesicle and the most useful type of EV for therapeutic purposes [75,105]. This is because exosomes have a diameter within the range of 30–200 nm, which is less than the diameter range of other EVs. Moreover, the exosomal molecular cargo is selected, meaning these nanosized vesicles are important nanocarriers of bioactive molecules of therapeutic interest [81]. For this reason, this review focuses on this class of EV.

## 4. Exosomes: From Biogenesis to Clinical Applications

### 4.1. Exosome Biogenesis

Exosomes are nanosized vesicles (30–200 nm) surrounded by a phospholipid membrane, containing cholesterol, sphingomyelin, ceramide, lipid rafts, and evolutionarily conserved markers such as tetraspanins (CD9, CD63, and CD81), heat shock proteins (HSP60, HSP70, and HSP90), major histocompatibility component (MHC) classes I and II, Alix, TSG101, lactadherin, and lysosome-associated membrane glycoprotein 2 [105].

Exosomes are secreted by all cell types and play a key role in cell-to-cell communication, being involved in both physiological and pathophysiological processes [106].

Exosomes are constitutively originated from late endosomes, which are formed by inward budding of the limited multivesicular body (MVB) [105,107,108].

MVBs and late endosomes are a subset of specialized endosomal compartments rich in intraluminal vesicles (ILVs) [106]. During the formation process of ILVs, certain proteins are incorporated into the invaginating membrane, while the cytosolic components are engulfed and enclosed within the ILV [108,109,110].

Multiple mechanisms explaining exosome biogenesis have been identified [75,105,106]. However, the endosomal sorting complex required for transport (ESCRT) remains the most studied mechanism involved in exosome biogenesis [111].

The ESCRT system consists of ESCRT-0, -I, -II, -III, and vacuolar protein sorting 4-vesicle trafficking 1 (VPS4-VTA1), as well as some accessory proteins such as the ALG-2-interacting protein X (ALIX) homodimer [111].

In eukaryotes, ESCRT-0 plays a key role in the recruitment of ESCRT-I to the endosomal membrane, which is crucial for the initiation of MVB-related cargo protein [111]. ESCRT-I (which in mammalians consists of TSG101, VPS28, VPS37, and HMVB12) is a heterodimer of about 20 nm, which interacts with ESCRT-0 and -II [111]. ESCRT-II is a Y-shaped heterodimer, with two subunits (EAP30 and EAP45 in mammalians) forming the base of the Y, which binds to the EAP20 subunit to form the Y arm [111]. ESCRT-II interacts with ESCRT-I, promoting the budding of the endosomal membrane to form the initial bud [111]. ESCRT-II also binds to ESCRT-III with high affinity, activating ESCRT-III, which is comprised of charged multivesicular body proteins (CHMP)-6, -4A, -4B, -4C, -3, -2A, and -2B. Once activated, ESCRT-III assembles into the endosomal membrane, in which it aggregates at the neck of the bud formed in the previous step and cleaves it, making it enter the endosomal compartment in the form of ILVs to form MVBs [111]. At the end of this process, energy is required to depolymerize ESCRT-III to allow it to enter the next cycle [111].

The ESCRT system is also related to the sorting process [112]. For this, accessory proteins, such as ALG-2-interacting protein X (ALIX) and tumor susceptibility gene 101 (TSG101), play a crucial role in cargo packing. ALIX not only encloses cargo to enter internalized vesicles, but also induces vesicle formation [113].

Upon maturation, the MVBs can be transported to plasma membrane via the cytoskeletal and microtubule network and undergo exocytosis post fusion with the cell surface, whereby the ILVs are secreted as exosomes [109,114]. The recruitment of exocytotic membranes (SNAREs) to the anchored MVB is the essential step, and then small GTPases (Rab27a and Rab27b) take part in the final release role [112]. SNARE proteins (SNAP23, syntaxin-4, and VAMP7) are involved in the release of exosomes [112]. However, it was demonstrated that some portion of exosomes remains attached to the parent cell surface, which are involved in signaling platforms for juxtracrine communication [115,116].

Alternatively, MVBs can also follow a degradation pathway either by direct fusion with lysosomes or by fusion with autophagosomes followed by lysosomes [106]. Although both secretory and degradatory MVB pathways coexist, the mechanism that influence these pathways remains unclear [106].

Once released into the extracellular space, exosomes interact with the ECM and recipient cells, serving as natural vehicles for the nano-delivery of nucleic acids, proteins, metabolites, and lipids [75,105,106].

### 4.2. Molecular Cargo

The enrichment of a particular set of molecules within the exosomes suggests the existence of specific sorting mechanisms that orchestrate the selective packaging of the RNAs and proteins [75,117].

For many years, this sorting mechanism remained unclear. However, evidence has demonstrated that that selective packaging of RNAs and proteins is governed by the endosomal sorting complex required for transport (ESCRT), which also contributes to exosome formation [75].

The ESCRT-mediated sorting is initiated by recognition and sequestration of ubiquitinated proteins to specific domains (Hrs FYVE domain with PtdIns3P—phosphatidylinositol 3-phosphate) of the endosomal membrane via ubiquitin-binding subunits of ESCRT-0 [108,118]. Next, the Hrs PSAP domain of the ESCRT-0 interacts with the subunit tumor susceptibility gene 101 (TSG101) of ESCRT-I [108,118]. ESCRT-I recruits the ESCRT-II proteins, which recruit and activate the ESCRT-III complex; in turn, this promotes the budding processes [108,118]. This occurs because the Snf7 protein of the ESCRT-III complex forms oligomeric assemblies, promoting vesicle budding [108,118]. Snf7 also recruits the Alix protein, stabilizing the ESCRT-III assembly [108,118]. Following cleaving the buds to form ILVs, the ESCRT-III complex separates from the MVB membrane with energy supplied by the sorting protein ATP Vps4 [75,108].

However, studies have showed the presence of ILVs within the lumen of MVBs in the ESCRT-depleted cells, indicating that the ESCRT-independent pathways for ILV formation exist [119,120].

In this sense, evidence suggests an alternative pathway for sorting exosomal cargo into MVBs in an ESCRT-independent manner, which seems to depend on raft-based microdomains for the lateral segregation of cargo within the endosomal membrane [108,119]. These microdomains are rich in sphingomyelinases, which are responsible for the formation of ceramides through the hydrolytic removal of the phosphocholine moiety [108,121]. These ceramides form a cone-shaped structure that causes spontaneous negative curvature of the endosomal membrane, thereby promoting domain-induced budding [108,121].

Moreover, proteins such as tetraspanins also participate in exosome biogenesis and protein loading. Tetraspanin-enriched microdomains (TEMs) are ubiquitous specialized membrane platforms for compartmentalizing of receptors and signaling proteins in the plasma membrane [108,122,123].

Thus, sorting mechanisms can select proteins and RNAs that will comprise the exosome content. For this reason, it is expected that exosomes derived from non-cancer cells and cancer cells engage in distinct activities in both physiology and pathophysiology [75].

### 4.3. Exosome Internalization in Recipient Cells

Exosomal internalization and intracellular trafficking are necessary to ensure that the exosomes cross the biological barriers and deliver the vesicular cargo components [112]. Cellular uptake of exosomes can occur via clathrin-mediated endocytosis, lipid raft-mediated endocytosis, heparan sulfate proteoglycans-dependent endocytosis, phagocytosis, or direct fusion with the plasma membrane [112]. However, the uptake of exosomes depends on the extracellular signal-regulated kinase, Hsp27 signaling, and lipid raft-mediated endocytosis [124].

Once internalized, exosomes can fuse with lysosomes/endosomes, releasing their cargo content in the cytoplasm, thereby transferring the cellular information to recipient cells [112], as shown in Figure 1.

### 4.4. Exosomes in Cancer Metastasis

Cumulative evidence has shown that both cancer-cell-derived and TME-cell-derived exosomes (also known as oncosomes) are involved in cancer progression, migration, invasion, and metastasis [75,125,126]. This is because exosomes carriers a plethora of bioactive molecules that can reprogram recipient cells.

In this sense, one of the most relevant discoveries of the exosome role in the oncogenic process was that of Hoshino et al. [127], who demonstrated that exosomal integrins (ITGs) have an important role in organ-specific metastasis in distant sites. This pioneering study showed that exosomal ITGα_6_β_4_ and ITGα_6_β_1_ are associated with lung metastasis, while ITGα_v_β_4_ is associated with liver metastasis and ITGβ_3_ is associated with brain metastasis. This study provided evidence that cancer-derived exosomes taken up by organ-specific cells prepare the pre-metastatic niche (PMN) [127].

In this regard, studies have shown that cancer-derived exosomes are involved in the inflammatory PMN formation in distant organs to promote metastasis [128,129,130]. This occurs through different immunomodulatory mechanisms, such as upregulation of pro-inflammatory cytokines [131] and nuclear factor kappa B (NF-κB)-dependent macrophage immunosuppression [132], which are both dependent on metabolic reprogramming.

In addition, cancer-derived exosomes increase lymphangiogenesis, as revisited by Yang et al. [95]. Lymph node metastasis is present in most cancers, negatively impacting the survival rate [16]. This is because patients with lymph node metastasis are prone to distant metastasis [95], since lymphangiogenesis in PMN can promote tumor metastasis [133]. In this sense, it was reported that exosomal long non-coding RNA (lncRNA) LNMAT2 could stimulate the tube formation and migration of human lymphatic endothelial cells (HLECs), promoting tumor lymphangiogenesis in bladder cancer [134]. Similar results were recently reported by Yao et al., who demonstrated that circular RNA (circ_0026611) contributes to lymphangiogenesis by reducing PROX1 acetylation and ubiquitination in HLECs [135]. Evidence describing the lymphangiogenic potential of cancer-derived exosomes has already been shown in studies based on melanoma [136], pancreatic cancer [137], and esophageal squamous cell carcinoma [138].

Cancer-derived exosomes also govern the organotropism, i.e., the capability of certain types of cancer to colonize and metastasize to specific organs under the control of a range of cellular and molecular programs [127,139,140].

In 1889, Dr. Stephen Paget proposed the “seed and soil” theory, describing that cancer cells (the “seeds”) metastasize to certain favorable organs (the “soils”) [141]. However, it was only in 2015 that Hoshino et al. [127] provided evidence that exosome-derived integrins drive the “seeds” for specific “soils”. Currently, it is well known that the exosomes serve as vehicles for the delivery of a plethora of naturally produced biomolecules, which simultaneously target multiple biological pathways. Among these molecules, the noncoding RNAs (ncRNAs) have gained attention, since these ncRNAs can regulate different steps of the oncogenic process, as exemplified in Table 2.

However, whether the exosomes can serve as nanocarriers for bioactive molecules involved in oncogenesis, or can serve as vehicles to transfer drugs to a specific organ [95], could be biotechnologically explored for delivery of chemotherapeutic drugs through cell-free therapy [75,112].

## 5. Mesenchymal Stromal/Stem Cells (MSCs) as a Source of Therapeutics Exosomes (MSC-Exo)

Exosomes are secreted by almost all type of cells, including cancer cells, and are found in different body fluids, such as blood plasma, urine, milk, saliva, and amniotic, bronchioalveolar, synovial, and ascites fluids [112]. However, the exosomal cargo content is entirely dependent on parental cells [112].

In this sense, while cancer-derived exosomes serve as useful biomarkers for the disease progression, and can be explored for liquid biopsies [157,158], mesenchymal stem/stromal cells (MSCs) have emerged as a useful source of exosomes for different therapeutic purposes, including for NDCs [75,159].

However, in recent years, many reviews have suggested that MSCs can be recruited for the TME, contributing to cancer progression and therapeutic resistance [160,161,162,163,164]. Studies have suggested this is because biomolecules produced and secreted by the TME promote the chemoattraction of MSCs [159].

Despite this concern regarding the use of these cells in patients with cancer, there is a lack of clinical evidence demonstrating that MSCs can cooperate with TME. Moreover, it should be noted that MSCs used for therapeutic purposes are a population of manufactured cells (in vitro) that do not exist in the body. Thus, it is not expected that these cells elicit the same properties of stem cells naturally found in human tissues.

Supporting this statement, we recently demonstrated that human immature dental pulp stem cells (hIDPSCs) produced at a large scale (as candidates for the treatment of neurodegenerative disorders [165]) do not engraft in pre-existing lung adenocarcinoma [166]. Moreover, because they share many characteristics with CSCs, it is empirically difficult to determine whether the pro-carcinogenic effects suggested by these reviews are related to MSC or CSC populations.

Nevertheless, this suggested pro-carcinogenic risk is even more reduced for MSC-derived exosomes, considering that these nanosized vesicles do not have replicative capability. Thus, MSCs can be considered a potential source of exosomes for clinical purposes, serving as NDCs. Reinforcing this hypothesis, studies have demonstrated that engineered and non-engineered MSC-derived exosomes possess different anti-cancer properties, as summarized in Table 3.

## 6. Chemotherapeutic Drug Load Techniques for Exosome-Based Drug Delivery System

Different strategies, including chemotherapeutics, have been used to load drugs into exosomes. Considering all cell types naturally produce these nanosized vesicles, one of the most straightforward drug-loading techniques involves incubating the cells with chemotherapeutic drugs of interest [184]. In this technique, the cells are set with a determined molar concentration of the chemotherapeutic drug. Next, the culture supernatants are collected and destined for exosome isolation (generally using ultracentrifugation). Following the exosomal characterization, a portion of the isolated exosomes is sonicated to release the chemotherapeutic drug. The released drug is measured using reverse-phase high-performance liquid chromatography (HPLC) [184,185,186].

Although relatively simple, this technique has several reported disadvantages: (i) Chemotherapeutic drugs tend to induce cytotoxicity and, therefore, lead to cell death in the cell population used as a source of exosomes. This cytotoxicity can increase the production of apoptotic vesicles, reducing both the efficacy and purity of the exosomal production for clinical purposes. (ii) Moreover, economically, this technique requires high quantities (concentrations) of the chemotherapeutic drugs to obtain exosomal preparations with the desirable drug concentration [184,186].

As an alternative to this technique, studies have demonstrated that electroporation (a method able to form pores on the membrane following an electrical signal stimulation) is a suitable strategy for loading drugs into the exosomes [187,188,189], including chemotherapeutics such as doxorubicin (DOX) [185,190,191]. Technically, this method is simple and offers the most advantages over the cultivation of exosome-producer cells with the target drug, avoiding cytotoxicity and, consequently, the production of apoptotic vesicles. However, this method can induce the exosomal lise. Furthermore, this method is not scalable and is not strategic for clinical purposes, since it requires large-scale production.

To avoid this technical limitation, studies have demonstrated that loading methods based on sonication, direct mixing, and incubation are suitable, especially for exosomal preparation of NDCs for chemotherapeutic drugs. In this sense, it was shown that Paclitaxel (PTX), an inferior aqueous soluble chemotherapeutic drug, can be loaded into exosomes via multiple cycles of sonication [192,193].

For hydrophilic chemotherapeutics, such as DOX, direct mixing and incubation are simple and efficient methods that can be used to load the drug into exosomes [194]. These methods reduce the concentration of chemotherapeutics needed to ensure an effective load and are economically preferable, especially for clinical purposes. Moreover, these methods can be easily scalable, allowing the manufacturer to have many modified exosomes.

### Exosomes as Nano-Drug Carriers (NDCs) for Chemotherapeutics

The scientific interest in exosomes started in the early 1970s, when numerous studies aimed to understand the biological function of these vesicles [103]. For almost a decade, exosomes were identified as a vehicle used to remove unnecessary molecules from the cells, like a garbage disposal [103,195]. However, in the 1990s, exosomes were identified in the cell-to-cell communication mechanism, playing a key role in both physiological and psychopathological processes [75,103,105,196]. Since then, the interest in these vesicles has increased yearly, as shown in Figure 2. This is because exosomes carrier a plethora of bioactive molecules that can simultaneously act in different ways in the pathophysiological process of a complex disease, such as cancer. In addition, exosomes have a reduced immunogenicity, serving as an attractive vehicle for the delivery of many drugs. Beside this, due to their small size and membrane composition, exosomes can cross biological membranes, including the blood–brain barrier [106]. Due to this capacity, these nanosized vesicles are potential candidates for cancer treatment.

The TME presents several barriers which hamper the delivery of therapeutic drugs to the site of action, such as vascular endothelial boundaries, the mononuclear phagocyte framework, low pH, low oxygenation, and high interstitial liquid [112]. These barriers result in considerable challenges to the delivery of a sufficient quantity of chemotherapeutic drugs to the target tumor site without harmful side-effects to healthy (normal) tissue [112]. In addition, metastatic cancer cells (particularly CSCs) overexpress ATP binding cassette (ABC) transporter, conferring chemoresistance to multiple drugs [75,197,198,199,200].

In this context, nano-drug delivery has emerged as a potential strategy to overcome the challenges related to chemotherapeutic drug failure and off-target toxicity [75,112,201,202,203]. This is because nano-drug carriers (NDCs) increase the drug solubility, bioavailability, and biodistribution. These NDCs enhance the permeability and retention effect by passing through leaky vasculature and poor lymphatic drainage of the tumor site [112,204,205]. Moreover, NDCs can encapsulate ionic, hydrophilic, and hydrophobic drugs [112].

Ideally, NDCs should be stable, nonimmunogenic, biodegradable, cost-effective, easy to fabricate, convenient for drug loading, and able to release their cargo at the targeted site of action [112]. Meeting these criteria, liposomes, micelles, dendrimers, antibodies, and polymeric materials are among the clinically approved NDCs [112]. However, opsonization of nanoparticles by the reticuloendothelial system (RES) is the main drawback of NDCs [112]. For this reason, studies examining the coating of the surface of nanoparticles with polyethylene glycol (PEGylation) have been performed to bypass the RES [112,206,207,208,209]. However, to date, no ideal NDC exists.

In this sense, a recent consensus has been reached for the use of exosomes as vehicles for chemotherapeutics drug delivery in place of synthetic NDCs [75,112,201,202,203]. This is because endogenous NDCs can deliver chemotherapeutic drugs with higher efficiency and lesser side-effects than synthetic NDCs [210,211]. This occurs because exosomes have favorable tumor homing properties and high stability [112]. The stability of the exosomes is the main advantage of these EVs, making these nanosized vesicles viable advanced therapy medicinal products (ATMPs). Altogether, due to these properties, the exosomes are suitable as NDCs for chemotherapeutics.

However, considering that the exosomes share a transcriptome signature with their origin cells, it is necessary to identify useful sources of exosomes for clinical purposes [75,105,196]. Although any cell type can serve as a source of exosomes, mesenchymal stromal/stem cells (MSCs, also known as signaling cells) can be considered an important source of exosomes for therapeutic purposes. This is because these cells can be easily isolated and expanded in vitro conditions [196,212,213,214]. These properties are necessary to produce these cells at a large scale for commercial purposes [215]. Moreover, the safety of MSCs has been previously demonstrated in numerous clinical trials for different diseases, as reported in several metanalysis [216,217,218,219,220,221,222].

## 7. Conclusions

Exosomes are naturally produced and secreted by all cell types, mediating cell-to-cell communication in physiological and pathophysiological conditions. In cancer, cumulative evidence has demonstrated that these nanosized vesicles transport nucleic acids (particularly coding and non-coding RNAs), proteins, and metabolites able to regulate and reprogram recipient cells. For this reason, it is no surprise that cancer-derived exosomes can govern multiple steps of the carcinogenic process, including the EMT and CSC formation. However, due to their reduced size (i.e., a diameter of less than 200 nm) and their capability to overcome the limitations imposed by the TME barriers, exosomes have emerged as a helpful candidate for the nano-delivery of chemotherapeutic drugs to the tumor. In this sense, the conditioned culture medium of MSCs, commonly discarded as a byproduct of the manufacturing process of cellular therapy, serves as an essential source of exosomes for clinical purposes in a new era of cancer therapy, known as cell-free therapy. However, concerns regarding the safety of MSCs for patients with cancer have limited the use of MSC-derived exosomes as NDCs for scientific purposes. Although relevant, this concern remains based on a few in vitro studies and reviews. In addition, the existence of a subpopulation of dedifferentiated cancer cells that share many characteristics with MSCs (known as CSCs) presents a technical difficulty, making it challenging to determine whether the pro-carcinogenic potential described by the literature is conferred by CSC or MSC populations. Nevertheless, recent studies have proved that exosomes derived from different MSC populations have anti-cancer properties. Although more efforts are needed to ensure the safety of MSC-derived exosomes as NDCs, to date, the available evidence suggests that these exosomes are potential candidates for the drug delivery of chemotherapeutics.

## Figures and Tables

**Figure 1 cells-12-02144-f001:**
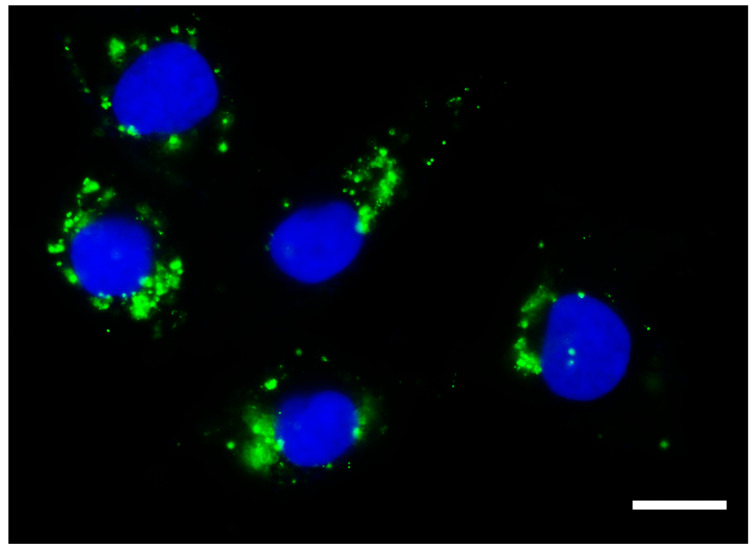
Exosomes taken up by cancer cells. Photomicrography obtained using high-content screening showing exosomes isolated from the conditioned culture medium of human immature dental pulp stem cells (stained with Vybrant DiO, in green) within the cytoplasm of a cancer cell line derived from metastatic anaplastic thyroid cancer (HTh83). Photomicrograph obtained 24 h after the addition of 50 µg/mL of exosomes. Scale bar 10 µm. Nucleus stained with Hoechst. Total magnification of 100×.

**Figure 2 cells-12-02144-f002:**
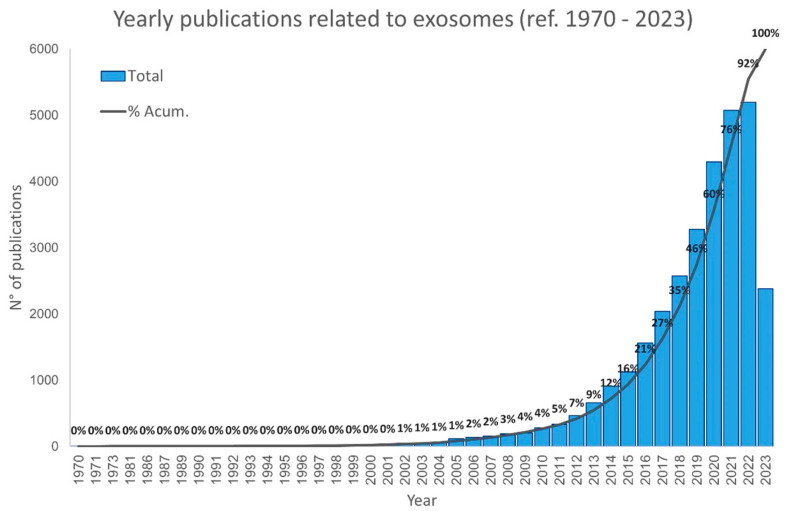
Number of publications involving exosomes since 1970. Data were obtained from PubMed. Bars show the total number of published articles involving exosomes per year, and the line shows the cumulative percentage of published articles. Graphics show that the number of articles involving exosomes has increased faster from 2015. Data collected to July 2023.

**Table 1 cells-12-02144-t001:** Extracellular classification based on biogenesis, concept, and size.

EV Category	Name	EV Class	Size (nm)	Markers	Biogenesis
Exosomes	Classical	Small	30–200	CD63^+^/CD9^+^/CD81^+^	MVE
	Non-classical	Small	30–200	CD63^+^/CD9^+^/CD81^−^	MVE
Microvesicle	Classical	Large	150–1000	Annexin A1, ARF6	PMS
	Oncosomes	Large	1000–10,000	Annexin A1, ARF6	PMS
	ARMM	Small	40–100	ARRDC1, TSG101	PMS
Apoptotic	Apoptotic body	Large	1000–5000	Annexin V, PS	Apoptosis
	Apoptotic vesicle	Small to large	100–1000	Annexin V, P5	Apoptosis
Autophagic	Autophagic EV	Small to large	40–1000	LC3B-PE, p62, dsDNA/histones	Amphisome
Stressed (Stressome)	Stressed EV	Small to large	40–1000	HSP90, HSPs	PMS
	Damaged EV	Small to large	40–1000	CD63^+^/CD9^+^/CD81^+^	PMS
Matrix vesicles	Matrix vesicles	Small to large	40–1000	Fibronectin, Proteoglycan	Matrix binding release

MVE—multivesicular endosome; ARMM—arrestin-domain-containing protein 1 (ARRDC1)-mediated microvesicle; PMS—plasma membrane shedding; Amphisome—Autophagosome-endosome fusion. Table adapted from Sheta et al. [104].

**Table 2 cells-12-02144-t002:** Noncoding RNAs identified in cancer-derived exosomes regulating several steps of the oncogenic process.

Mediated Effects	Cancer Type	microRNAs	Reference
Progression	Breast cancer	miR-7641	Shen et al. [142]
	Breast cancer	miR-1304–3p	Zhao et al. [143]
	Breast cancer	miR-500a-5p	Chen et al. [144]
	Liposarcoma	miR-25-3p, and miR-92a-3p	Casadei et al. [145]
	Leukemia	miR-21	Li et al. [146] Agatha et al. [147]
	Colorectal cancer	miR-193a and let-7g	Cho et al. [148]
Migration	Breast cancer	miR-7641	Shen et al. [142]
	Hepatocellular carcinoma	miR-140-3p, miR-30d-5p, miR-29b-3p, miR-130b-3p, miR-330-5p and miR-296-3p	Yu et al. [149]
	Ewing sarcoma	miR-34a	Ventura et al. [150]
	Esophageal carcinoma	miR-21	Ragusa et al. [151]
Immune evasion	Epithelial ovarian cancer	miR-940, miR-21-3p, miR-125b-5p	Chen et al. [152]
	Colorectal cancer	miR-203, miR-145, miR-934, miR-1246, miR-25-5p, mirR-130b-3p, miR-425-5p, miR-21-5p	Wadhankar et al. [153]
Differentiation	Ewing sarcoma	mi-34a	Ventura et al. [150]
Chemoresistance	Breast cancer	miR-155	Santos et al. [154]
	Epithelial ovarian cancer	miR-223	Zhu et al. [155]
	Colorectal cancer	miR-208b	Ning et al. [156]

**Table 3 cells-12-02144-t003:** Anti-cancer properties identified in both engineered and non-engineered MSC-Exos derived from different types of MSCs.

MSC	Properties	Cancer Type	Reference
Adipocyte (AMSC)	Increase of chemosensitivity	Breast cancer	Jia et al. [167]
	Anti-proliferative	Ovarian cancer	Reza et al. [168]
	Anti-proliferative	Bladder cancer	Liu et al. [169]
	Anti-invasive		
	Anti-migration		
	Increase of chemosensitivity	Hepatocellular carcinoma	Lou et al. [170]
	Anti-proliferative	Glioblastoma multiforme	Pastorakova et al. [171]
Bone marrow (BMMSC)	Anti-angiogenic	Breast cancer	Lee et al. [172] Pakravan et al. [173]
	Anti-proliferative	Acute myeloid leukemia (AML)	Zhang et al. [174]
	Increase the delivery of Paclitaxel	Pancreatic adenocarcinoma	Pascucci et al. [175]
	Increase the delivery of Paclitaxel	Breast cancer	Khalimitu et al. [176]
	Increase the delivery of Doxorubicin	Neuroblastoma	Li et al. [177]
	Anti-proliferative	Glioblastoma multiforme	Pastorakova et al. [171]
Dental pulp (DPMSC)	Anti-proliferative	Glioblastoma multiforme	Pastorakova et al. [171]
	Increase the delivery of Paclitaxel	Breast cancer	Salehi et al. [178]
	Nano-delivery through intranasal route	Glioblastoma multiforme	Altanerova et al. [179]
Human umbilical cord (hUCMSC)	Anti-proliferative Pro-apoptotic	Prostate	Takahara et al. [180]
	Enhance radiotherapy-induced death	Melanoma	Farias et al. [181]
	Pro-apoptotic	Chronic myelogenous leukemia	Liu et al. [182]
	Anti-proliferative	Glioblastoma multiforme	Pastorakova et al. [171]
	Anti-proliferative	Breast cancer	Yuan et al. [183]

## Data Availability

Not applicable.

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
