# Peer review of "Exosomes as a Nano-Carrier for Chemotherapeutics: A New Era of Oncology"

_cells, 2023, doi:10.3390/cells12172144_

Round 1

Reviewer 1 Report

The theme is new, the citation is relative fresh, however, the arrangement of structure has a major problem, please consider the following advice:

1.     I noticed there’s a similar theme of another review article (Role of Exosomes for Delivery of Chemotherapeutic Drugs. Volume 38, Issue 5, 2021, pp. 53-97. DOI: 10.1615/CritRevTherDrugCarrierSyst.2021036301), which is also cited in your manuscript, could you please tell what’s the novelty in your manuscript comparing with that one?

2.     The title makes readers confused that they would think makybe exosome is just a good carrier for nano-drug, but actually it’s not. So, I’d suggest revising it as “Exosomes as a nano-carry of drug for chemotherapeutics: A new era of oncology” or “Exosomes as a nano-carry for chemotherapeutics: A new era of oncology”, FYI. Please also consider other parts with same or similar statements (nano-drug carry/carrier or nano drug carry/carrier).

3.     In keywords part, MSC-Exo is repeated from Mesenchymal Stem Cell-derived exosomes, please delete it.

4.     As background, Part 1 and part 2 have no straight link with exosome, please combine the two parts together and shorten them as possible. You have 14 pages of text, but this minor background knowledge occupies 5 pages, that’s too much!

5.     In the last paragraph of part 3, please explain why exosome stands out from extracellular vesicles.

6.     5 Chemotherapeutic drug load techniques for exosome-based drug delivery system should be 6 Chemotherapeutic drug load techniques for exosome-based drug delivery system, and 4.5 Exosomes as nano-drug carriers (NDCs) for chemotherapeutics should be added into this part to make the structure rational. Also, 6. Conclusions should be 7. Conclusions.

7.     Line 604, drugs can be deleted.

Author Response

Dear Reviewer,

We would like to thank for your comments and suggestions. We hope that we can respond to all points with this letter.

#Reviewer 1

The theme is new, the citation is relative fresh, however, the arrangement of structure has a major problem, please consider the following advice:

  1. I noticed there’s a similar theme of another review article (Role of Exosomes for Delivery of Chemotherapeutic Drugs. Volume 38, Issue 5, 2021, pp. 53-97. DOI: 10.1615/CritRevTherDrugCarrierSyst.2021036301), which is also cited in your manuscript, could you please tell what’s the novelty in your manuscript comparing with that one?

We thank you for the comparison of our manuscript with the standout published review. As a review, the main goal of our work is not providing novelty but summarize relevant discoveries that can support the use of exosome-based cell-free therapy in oncology. Comparing our manuscript with the standout review, please, note the in the work “Exosomes for Delivery of Chemotherapeutic Drugs” the authors focused on the methods to isolate exosomes, theme which has been constantly published by numerous authors. However, we focus on summarize the therapeutic challenges which need to be overcome, bring an overview about the metastatic process (section 2). In this sense, we brought a collection of evidence showing how the use of exosomes can help to overcome the barriers that naturally difficult the treatment of patients with metastatic disease (section 4.4). Moreover, we contribute with a section describing the application of mesenchymal stem cells as an important source of exosomes, theme which is not explored in the work “Exosomes for Delivery of Chemotherapeutic Drugs”. We hope that these differences can be sufficient to demonstrate the novelty of our review.

  1. The title makes readers confused that they would think makybe exosome is just a good carrier for nano-drug, but actually it’s not. So, I’d suggest revising it as “Exosomes as a nano-carry of drug for chemotherapeutics: A new era of oncology” or “Exosomes as a nano-carry for chemotherapeutics: A new era of oncology”, FYI. Please also consider other parts with same or similar statements (nano-drug carry/carrier or nano drug carry/carrier).

We thank you this correction and apologize for the misunderstood promoted by the title. As suggested, we modified the title to “Exosomes as a nano-carry for chemotherapeutics: A new era of oncology”.

  1. In keywords part, MSC-Exo is repeated from Mesenchymal Stem Cell-derived exosomes, please delete it.

We apologize for this mistake. Keywords part was modified and the term “mesenchymal stem cell-derived exosomes” was replaced to MSC.

  1. As background, Part 1 and part 2 have no straight link with exosome, please combine the two parts together and shorten them as possible. You have 14 pages of text, but this minor background knowledge occupies 5 pages, that’s too much!

We thank you for this comment. However, this is the main difference of our review to other reviews already published. We aim to bring an overview about the oncogenic process, with focus on metastatic process. Other reviews only claim the application of exosomes as nanocarriers for drug delivery, discussing the biogenesis and the methods for isolation these nanosized vesicles without any contextualization with the oncogenic process. For this reason, respectably, we recognized the importance to provide the details about the oncogenic process which are described in sections 1 and 2.

  1. In the last paragraph of part 3, please explain why exosome stands out from extracellular vesicles.

As requested, we introduce the following statement to justify the reason for which the exosomes are the most studies class of extracellular vesicles: “This is because, exosomes have a diameter within the range of 30-200 nm, which beside reduced when compared to the range of diameter of another EVs. Moreover, the exosomal molecular cargo is selected, making of these nanosized vesicles important nanocarriers of bioactive molecules of therapeutic interest [83]”.

  1. 5 Chemotherapeutic drug load techniques for exosome-based drug delivery system should be 6 Chemotherapeutic drug load techniques for exosome-based drug delivery system, and 4.5 Exosomes as nano-drug carriers (NDCs) for chemotherapeutics should be added into this part to make the structure rational. Also, 6. Conclusions should be 7. Conclusions.

We thank you for this observation. The section numbers were corrected, and the section 4.5 was added to the section 6 (now section 6.1) as suggested.

  1. Line 604, drugs can be deleted.

As requested, the word “drugs” was removed.

Reviewer 2 Report

The manuscript by Pinheiro Araldi and colleagues reviews the use of exosomes as drug-delivery systems in cancer. The authors provide first an introduction on cancer in general and the challenges that cancer metastasis represents. Finally, the authors introduce extracellular vesicles and exosomes, their biogenesis, sources and their applications as drug-delivery systems. The authors address a highly relevant topic. Although the introduction parts are exhaustive enough, the authors can elaborate more on the different studies using EVs as drug-delivery systems in vivo. In this regard, it would be very helpful if a table describing all these studies (model, drug, upload system) can be provided. In addition, the following aspects need to be addressed before publication:

1. Lines 30-32. While the first part of the statement refers to cancer incidence in 2022, the reference (Ref. 3) dates from 2020. Please, reformulate. Alternatively, Ref. 3 needs to be replaced for a more recent one. 

2. Reference 2 and Reference 6 are the same. The same applies to references 8 and 9 and to references 12, 15 and 43. References 75 and 76 are also the same. References 101 and 102 and references 103 and 104 are also repeated. 

3. Lines 106-107. The sentence needs to be checked and reformulated. 

4. In general, section 2 should be summarized. For example, in lines 107-118, the detailed description of oxidative glucose metabolism under physiological conditions seems to be rather beyond the scope of the section and the manuscript. I recommend summarizing this part and highlighting the differences between metabolism in healthy and in cancer cells.

5. Lines 168-176 partially repeat information described in lines 106-129.

6. Line 203. The citation of Ref. 79 is not fully appropriate in this part of the manuscript. The original article does not describe any aspect of EMT involving Beta-catenin signaling. 

7. Lines 216-218. The sentence could be misleading. In fact, MDR1 is only one of the ABC transporters expressed in CSCs and cancer cells in general. Other relevant ABC transporters in CSCs (e.g., BCRP/ABCG2) do not fall under the MDR denomination. 

8. References 110 and 114 are the same, references 111 and 113 as well. References 117 and 120 are the same. References 121 and 126 are the same. References 127 and 128 are the same. 

9. Please add references to support the information in table 1. 

10. Also, in table 1, there is a typo in “ARMN”. It should be “ARMM”, as properly defined in the table footer. 

11. The sentence in lines 310-312 should be checked and reformulated. 

12. In table 2, the authors postulate an association between miR-21 and leukemia progression. While, as demonstrated by Li et al. (Ref 164), the diagnostic potential of miR-21 in serum EVs is clear, the mechanistic role of miR-21 promoting leukemia progression is not fully demonstrated in that publication. A reference clearly linking miR-21 and progression at cellular level should be provided. 

13. Table 2 (Ewing Sarcoma). The miRNA mediating the effects is miR-34a, instead of miR-99. Please, check. 

14. Lines 449-455. This information is rather general and should be moved to the beginning of section 3. 

15. Table 3. There is a typo in “Breas cancer”. 

16. Table 3. 3rd row within BMMSC. The information on cancer type needs to be completed (“Pancreatic adenocarcinoma” instead of “Adenocarcinoma”)

17. Lines 556-562. References need to be added.

Minor editing required. Manuscript should be checked for typos. 

Author Response

Dear Reviewer,

We would like to thank you for your comments and suggestions. We hope that we can respond to all points with this letter.

he manuscript by Pinheiro Araldi and colleagues reviews the use of exosomes as drug-delivery systems in cancer. The authors provide first an introduction on cancer in general and the challenges that cancer metastasis represents. Finally, the authors introduce extracellular vesicles and exosomes, their biogenesis, sources and their applications as drug-delivery systems. The authors address a highly relevant topic. Although the introduction parts are exhaustive enough, the authors can elaborate more on the different studies using EVs as drug-delivery systems in vivo. In this regard, it would be very helpful if a table describing all these studies (model, drug, upload system) can be provided. In addition, the following aspects need to be addressed before publication:

  1. Lines 30-32. While the first part of the statement refers to cancer incidence in 2022, the reference (Ref. 3) dates from 2020. Please, reformulate. Alternatively, Ref. 3 needs to be replaced for a more recent one. 

We apologize for the mistake in relation to the reference year. For this reason, we modified 2022 to 2020 and rewrote the sentence as “According to the GLOBOCAN, about 19.3 million new cases of cancer were registered globally in 2020 [3], which represents about 5,250 new cancer cases each day only in United States for this same year [4]. The tabulation and graphical visualization of the GLOBOCAN data can be accessed via the Global Cancer Observatory (GCO) (http://gco.iarc.fr)”.

  1. Reference 2 and Reference 6 are the same. The same applies to references 8 and 9 and to references 12, 15 and 43. References 75 and 76 are also the same. References 101 and 102 and references 103 and 104 are also repeated. 

We apologize for the repeated references. They were corrected in the main text.

  1. Lines 106-107. The sentence needs to be checked and reformulated. 

We thank you for this observation. The terms Hif1a activation was inappropriately present in the beginning of the sentence. We excluded this term, modifying the sentence to “Cancer demands ATP to fulfill the biosynthesis need for proliferation [26]. Under physiological conditions, normal cells produce ATP by the oxidative metabolism. Using a series of coordinated enzymatic reactions, these cells convert the glucose in pyruvate in a cytoplasmic reaction (glycolysis) that occurs independently of the oxygen”

  1. In general, section 2 should be summarized. For example, in lines 107-118, the detailed description of oxidative glucose metabolism under physiological conditions seems to be rather beyond the scope of the section and the manuscript. I recommend summarizing this part and highlighting the differences between metabolism in healthy and in cancer cells.

As suggested, the sentences were rewritten aiming to summarize the energy supply process.

  1. Lines 168-176 partially repeat information described in lines 106-129.

We thank you for this comment. However, not that in lines 106-129 is described the physiological condition in which glucose is used to generate ATP under oxidative conditions (i.e., in the presence of oxygen). By contrast, in lines 168-176 is described the metabolic reprograming observed in cancer cells, which is known as Warburg effect. For this reason, we consider important to maintain the text structure to avoid any misinterpretation in relation to normal and cancer cells metabolism.

  1. Line 203. The citation of Ref. 79 is not fully appropriate in this part of the manuscript. The original article does not describe any aspect of EMT involving Beta-catenin signaling. 

Reference 79 was replaced by DOI: 10.3390/cancers10120491

  1. Lines 216-218. The sentence could be misleading. In fact, MDR1 is only one of the ABC transporters expressed in CSCs and cancer cells in general. Other relevant ABC transporters in CSCs (e.g., BCRP/ABCG2) do not fall under the MDR denomination. 

In order to avoid any misinterpretation, the sentence was rewritten to “In addition, as well as observed in stem cell populations, CSCs also express high levels of ATP-binding cassette (ABC) transporters [83] and DNA damage response genes [81,84].”

  1. References 110 and 114 are the same, references 111 and 113 as well. References 117 and 120 are the same. References 121 and 126 are the same. References 127 and 128 are the same. 

We apologize for the repeated references. They were corrected in the main text.

  1. Please add references to support the information in table 1. 

Reference was added in the footnote of the table.

  1. Also, in table 1, there is a typo in “ARMN”. It should be “ARMM”, as properly defined in the table footer. 

We apologize for the typo, it was corrected.

  1. The sentence in lines 310-312 should be checked and reformulated. 

The sentence was rewritten, excluding the duplicate term “the neck”

  1. In table 2, the authors postulate an association between miR-21 and leukemia progression. While, as demonstrated by Li et al. (Ref 164), the diagnostic potential of miR-21 in serum EVs is clear, the mechanistic role of miR-21 promoting leukemia progression is not fully demonstrated in that publication. A reference clearly linking miR-21 and progression at cellular level should be provided. 

We thank you for this comment. Evidence supporting the mechanistic role of miR-121 in leukemia was provided by the added reference DOI  10.3390/cells9092053

  1. Table 2 (Ewing Sarcoma). The miRNA mediating the effects is miR-34a, instead of miR-99. Please, check. 

We apologize for the mistake. The miR was modified.

  1. Lines 449-455. This information is rather general and should be moved to the beginning of section 3. 

We thank you for this suggestion. To improve the comprehension, we replaced the section 4.5 to 6.1.

  1. Table 3. There is a typo in “Breas cancer”. 

We thank you for this observation. The typo was corrected.

  1. Table 3. 3rd row within BMMSC. The information on cancer type needs to be completed (“Pancreatic adenocarcinoma” instead of “Adenocarcinoma”)

The cancer type “adenocarcinoma”, which was missing the cancer type, was changed to pancreatic adenocarcinoma as requested.

  1. Lines 556-562. References need to be added.

As requested, the references were added.

Round 2

Reviewer 2 Report

The authors have answered to all my concerns and provided an improved version of the manuscript.